# Allogeneic Hematopoietic Stem Cell Transplantation for Adults with Sickle Cell Disease

**DOI:** 10.3390/jcm8101565

**Published:** 2019-10-01

**Authors:** Santosh L. Saraf, Damiano Rondelli

**Affiliations:** Division of Hematology & Oncology, Department of Medicine, University of Illinois at Chicago, Chicago, IL 60612, USA; drond@uic.edu

**Keywords:** sickle cell disease, transplantation

## Abstract

Sickle cell disease (SCD) is an inherited red blood cell disorder that leads to substantial morbidity and early mortality. Acute and chronic SCD-related complications increase with older age, and therapies are urgently needed to treat adults. Allogeneic hematopoietic stem cell transplantation (HSCT) is a curative therapy, but has been used less frequently in adults compared to children. This is, in part, due to (1) greater chronic organ damage, limiting tolerability to myeloablative conditioning regimens, (2) a higher rate of HSCT-related complications in adults versus children with SCD, and (3) limited coverage by public and private health insurance. Newer approaches using nonmyeloablative and reduced-intensity conditioning HSCT regimens have demonstrated better safety and tolerability, with high rates of stable engraftment in SCD adults. This review will focus on the impacts of HSCT, using more contemporary approaches to SCD-related complications in adults.

## 1. Morbidity and Mortality of Sickle Cell Disease

Sickle cell disease (SCD) is among the most common monogenetic disorders, affecting approximately 25 million people worldwide [1]. Sickle cell disease is due to homozygous inheritance of the hemoglobin S mutation (Hb SS) or compound heterozygous inheritance of the Hb S mutation with another β-globin chain abnormality, such as Hb C (Hb SC) or β-thalassemia (Hb Sβ^+^-thalassemia, Hb Sβ^0^-thalassemia). Under deoxygenated conditions, Hb S polymerizes, resulting in vaso-occlusion, chronic red blood cell hemolysis, acute and chronic organ complications, and reduced overall survival [2]. Allogeneic hematopoietic stem cell transplantation (HSCT) is a curative therapy that has been applied more frequently in children than adults with SCD [3].

### 1.1. Contemporary Survival Patterns in Sickle Cell Disease

The Comprehensive Study of Sickle Cell Disease (CSSCD) cohort study demonstrated that the median survival for men and women was 42 and 48 years in Hb SS SCD and 60 and 68 years in Hb SC SCD, respectively [4]. More recent cohorts have demonstrated that, with advances in care, such as newborn screening, penicillin prophylaxis, and effective vaccinations, the proportion of children surviving into adulthood has increased [5,6]. However, overall median survival remains substantially lower in SCD than in the general population. Based on mortality data from the Centers for Disease Control, mortality was highest in men 35–44 years of age and in women 45–54 years of age. The predominant causes of death were related to organ injury and included cardiac (32%), pulmonary (28%), renal (16%), and neurologic (12%) disease. Several other cohorts have demonstrated median survivals ranging between 38 and 67 years in Hb SS patients and 55 and 66 years in Hb SC or Hb Sβ^+^-thalassemia patients [7,8,9,10,11]. In a pooled analysis of SCD patients from two academic SCD centers in the USA, the median overall survival for Hb SS/Hb Sβ^0^-thalassemia/Hb SD and for Hb SC/HbSβ^+^-thalassemia was 48 and 55 years, respectively. These estimates are relatively unchanged from the median survivals observed in the CSSCD cohort approximately 25 years ago, highlighting the need for better interventions in SCD. Although improvements in some causes of mortality, such as from pneumococcal sepsis, have been observed, acute and chronic organ complications remain independent predictors for early mortality in SCD [4,10,11,12,13,14,15,16].

### 1.2. Acute Complications in Adults with Sickle Cell Disease

Acute pain is a hallmark feature of SCD that is a consequence of impaired oxygen delivery and ischemia‒reperfusion injury from vaso-occlusion. In a recent multicenter, prospective study of adults with SCD, 54% of patients with Hb SS and 46% with Hb SC had three or more vaso-occlusive crises (VOC) requiring medical attention [17]. In addition to being commonly observed, frequent VOC are associated with a 2.1–2.7-fold greater risk of early mortality compared to those with less frequent VOC [9,10]. Acute chest syndrome is defined by a new pulmonary infiltrate on chest radiograph, accompanied by fever and/or respiratory symptoms. It is the second most common cause of hospitalization, after VOC, and the leading cause of death in hospitalized SCD patients [4]. The occurrence of acute chest syndrome is associated with a 1.8–2.4-fold greater risk of mortality compared to those who do not develop an acute chest syndrome event [12,13]. 

The overall lifetime risk of a clinically overt or silent stroke is estimated at up to 30% and 43%, respectively, in patients with SCD [18,19]. In children, primary stroke prevention can be achieved by red blood cell transfusion therapy in those with elevated transcranial Doppler velocities [20]. In adults, stroke management is focused on secondary stroke prevention as cutoff values for stroke risk using transcranial Doppler velocities are less clear [21]. If untreated, SCD patients with an overt stroke have an approximately 50% risk of a recurrent stroke within the first two years after the initial event and up to a 66% risk on longer follow-up [22,23]. Red blood cell transfusion therapy, targeting a goal Hb concentration of 10 g/dL and Hb S < 30%, improves the risk of recurrence to 18% for an overt stroke and 28% for new silent infarcts [24]. The occurrence of stroke is also an independent risk factor for a 2-fold greater risk of death [12].

### 1.3. Cardiovascular Complications in Sickle Cell Disease

Chronic cardiovascular disease has been increasingly recognized as a leading cause of morbidity and mortality in adults with SCD. In one adult SCD cohort, 26% of deaths were attributed to cardiac causes [16]. The tricuspid regurgitant jet velocity (TRJV) is a noninvasive marker of systolic pulmonary artery blood pressure obtained by transthoracic echocardiography that is elevated in SCD patients with pulmonary hypertension or diastolic dysfunction. In a prospective cohort of SCD adults, a TRJV ≥ 2.5 m/s was observed in 32% of patients and was associated with a 10.1-fold increased risk of death [25]. Several other cohorts have reproduced the association of an elevated TRJV ≥ 2.5‒2.7 m/s as an independent predictor for mortality [26,27,28]. The plasma N-terminal-pro-brain natriuretic peptide (NT-proBNP) is a biomarker of cardiac wall stress that is released from myocytes in the left and right ventricles under conditions of pressure overload. An NT-proBNP > 160 pg/mL is observed in 24–30% of SCD adults and is associated with a 2.9–5.1-fold greater risk of death [29,30]. 

### 1.4. Health-Related Quality of Life

The impact of worsening acute and chronic SCD-related complications in adulthood leads to a reduced health-related quality of life (HRQOL). In a longitudinal cohort study of SCD adults, HRQOL was substantially lower in men and women compared to national norms for all SF-36 subscales, with the exception of the mental health scale [31]. Furthermore, when comparing the SF-36 subscale scores between adults with SCD versus age-matched patients with end-stage renal disease or cystic fibrosis, SCD patients scored significantly lower in several domains. This included lower scores for physical function, physical role, bodily pain, general health, and vitality. 

### 1.5. Healthcare Utilization and Healthcare Costs

In parallel with the profound morbidity, SCD is associated with an increased burden on the healthcare system. It is currently estimated that medical care for treating SCD costs approximately $1.1 billion/year, with 81% of the costs being attributed to inpatient hospital care [32]. The total healthcare-related cost increases with age and is estimated between $34,266 and $231,050 per patient per year in adults with SCD [32,33]. 

## 2. Currently Accepted Indications for Allogeneic hematopoietic stem cell transplantation (HSCT) in Sickle Cell Disease

Based on the acute and chronic complications that lead to greater morbidity and mortality in SCD, several guidelines have been proposed for when to consider allogeneic HSCT [34,35,36,37]. Most of the earlier guidelines focused on acute complications (e.g., stroke, recurrent hospitalizations for VOC or acute chest syndrome), chronic organ complications (e.g., impaired neurocognitive function with abnormal brain MRI scan, early stages of sickle lung disease, nephropathy, retinopathy, or avascular necrosis of multiple joints), or red blood cell alloimmunization during long-term transfusion therapy [37]. More recent guidelines have included an elevated TRJV, removed sickle lung disease, and bilateral proliferative retinopathy as indications for HSCT. Table 1 provides criteria that have been proposed for HSCT using either HLA-matched related or alternative donors in SCD [34,35,36,37]. Due to the concern of greater HSCT-related complications when using an alternative donor as the graft source, guidelines recommend that hydroxyurea or chronic red blood cell transfusion therapy be tried prior to HSCT using an unrelated or haploidentical donor source [34].

The indications for HSCT should be carefully considered in the context of current and emerging therapies for SCD. Hydroxyurea is a ribonucleotide reductase inhibitor that increases hemoglobin F%. Hydroxyurea may also improve vascular endothelial damage in SCD by increasing nitric oxide generation by red blood cells and by lowering white blood cell and reticulocyte counts [38]. In the landmark Multicenter Study of Hydroxyurea, adult SCD patients randomized to hydroxyurea had significantly lower rates of VOC, acute chest syndrome, and red blood cell transfusions [39]. Several cohorts have demonstrated that long-term use of hydroxyurea may lead to improved survival patterns, particularly in those that have a high hemoglobin F response, although its effects on chronic organ damage are less clear [13,40,41]. l-glutamine is an essential amino acid that is required to synthesize nicotinamide adenine dinucloeotide (NAD), a redox cofactor in red blood cells that is deficient in SCD. In a multicenter, randomized, placebo-controlled phase 3 study, l-glutamine led to a modest reduction in VOC, with a median of 3.0 VOC in the l-glutamine group versus 4.0 VOC in the placebo group during a 48-week treatment period [42]. Newer therapies targeting P-selectin (crizanlizumab) or inhibiting hemoglobin S polymerization (voxelotor) have recently been tested in randomized, placebo-controlled, double-blind studies in adults with SCD. Crizanlizumab (5 mg/kg) led to a 45% decline in VOC rates compared to the placebo (2.98 vs. 1.63, respectively) over 52 weeks of therapy [43]. Voxelotor (1500 mg/day) improved hemolysis, reflected by improvements in mean change in hemoglobin concentrations (+1.1 g/dL vs. −0.1g/dL) and indirect bilirubin (−19.9% vs. +4.5%), compared to the placebo [44]. 

## 3. Allogeneic HSCT in Adults with Sickle Cell Disease

### 3.1. Global Experience

A combined analysis assessed outcomes in 1000 SCD patients undergoing allogeneic HSCT between 1986–2013 using data from the Center for International Blood and Marrow Transplant Research (CIBMTR), European Society for Blood and Marrow Transplantation (EBMT), and Eurocord databases [3]. In this study, only 154 (15%) of the SCD patients were adults (median age: 19 years, range: 16–54 years); 73% received a myeloablative conditioning regimen and 26% received a reduced-intensity conditioning regimen. Overall survival and event-free survival were significantly lower in patients 16 years or older (81% for each) versus those younger than 16 years (95% and 93%, respectively). The five-year probability of graft versus host disease (GVHD [22] was higher in patients 16 years or older (23%) versus those younger than 16 years (14%). Furthermore, for every one-year increment in age at transplantation, there was a 4% increase in the hazard ratio for acute GVHD, a 9% increased risk of graft failure, and a 10% increased risk of death. These results highlight the toxicity of myeloablative conditioning regimens in adult SCD patients, but did not include more recent series of adults undergoing HSCT using less intensive conditioning regimens.

### 3.2. Myeloablative Conditioning Regimens

Most of the earlier HSCT regimens in SCD used myeloablative doses of busulfan and cyclophosphamide and were predominantly performed in children [37,45,46,47]. One report on 15 young SCD adults with a median age of 19 years (range: 16–27) that underwent HSCT using a busulfan (485 mg/m^2^), cyclophosphamide (200 mg/kg), and anti-thymocyte globulin (ATG) (20 mg/kg) conditioning regimen and HLA-matched sibling donors demonstrated an overall survival and disease-free survival of 93% [48] Acute (≥grade 2) and chronic GVHD occurred in seven (47%) and two (14%) of the patients, respectively. Two of these patients had mixed chimerism (>75%), and all 14 living patients were tapered off immunosuppression.

### 3.3. Reduced-Intensity Conditioning Regimens

Reduced-intensity conditioning regimens have been studied in SCD adults in order to improve the tolerability for allogeneic HSCT. In one study, two SCD adults (40 and 56 years old) underwent allogeneic HSCT using a fludarabine‒melphalan conditioning regimen with ATG from a matched related donor [49]. Both patients tolerated the conditioning regimen reasonably, demonstrated stable engraftment, and had subsequent episodes of VOC. Unfortunately, the first patient developed chronic GVHD of the lung and passed away on day +335 and the second patient developed acute GVHD of the gut and liver and passed away on day +147.

More recent studies using fludarabine and lower doses of busulfan in SCD adults have demonstrated improvements in event-free and overall survival and in lower rates of acute and chronic GVHD, comparable to what has been observed in some pediatric cohorts (Table 2). In a retrospective study of 20 SCD adults (mean age of 33 years) undergoing HSCT using HLA-matched related donors at the University of Baskent, all 20 had stable engraftment at the last follow-up and 11 of 12 patients who completed one year of follow-up were off all immunosuppression [50]. In the prospective, multicenter STRIDE study of 22 adolescents and young adults (median age: 23 years) undergoing HSCT using either HLA-matched related or unrelated donors, two patients had secondary graft failures and two deaths were observed [35]. 

### 3.4. Nonmyeloablative Conditioning Regimens

Prior studies demonstrated that stable mixed chimerism, defined as a state wherein the lymphohematopoietic system is composed of a mixture of recipient and donor-derived blood cells, is sufficient to improve laboratory parameters and prevent SCD-related complications [51] Stable mixed chimerism is also associated with an induction of tolerance that leads to lower rates of GVHD compared to full donor chimerism [52]. A recent study [53] and a mathematical model [54] demonstrated that at least 20% myeloid donor chimerism is necessary to improve the SCD phenotype. In lethally irradiated mice transplanted with varying ratios of donor sickle and normal marrow, a myeloid chimerism >25% from normal marrow was sufficient to restore the Hb content to >90% normal, consistent with what has been observed in clinical studies [55]. 

Applying this concept, a nonmyeloablative approach using alemtuzumab/total body irradiation to achieve a stable mixed chimerism without substantial HSCT-related toxicity was developed at the National Institutes of Health (NIH) (Table 3). Initial reports demonstrated stable engraftment in nine out of 10 SCD adults, with no acute or chronic GVHD [56]. In a follow-up study of 30 SCD adults (median age: 29 years), four had secondary graft failure and there was one death [57]. No patients developed acute or chronic GVHD and 15 discontinued immunosuppression. Comparable outcomes were observed in another prospective study from the University of Illinois at Chicago, utilizing the same approach [58]. In this cohort of 13 SCD adults (median age: 30 years), one patient had secondary graft failure and no deaths or GVHD were observed. This same regimen has also been investigated at the King Saud Bin Abdulaziz University for Health Sciences in 17 adults with SCD (median age: 24 years). In this study, one secondary graft failure but no deaths or GVHD was observed [59]. 

### 3.5. Alternative Donor Approaches

The limited availability of an HLA-matched donor is a major barrier to HSCT in SCD. In one study, only 24 of 112 (21%) adult SCD patients meeting the eligibility criteria for HSCT had an HLA-matched related donor [56]. Similarly low rates of HLA-matched related donor availability have been observed in other adult and pediatric SCD cohorts [58,60,61]. Furthermore, in the National Marrow Donor Program donor registry, it is estimated that people of African descent have only a 16% probability of finding an HLA-matched donor [62]. The discovery that posttransplant immunosuppression with high doses of cyclophosphamide (PTCy) allows patients to engraft haploidentical donor cells without increasing the risk of GHVD has led to the rapid implementation of haploidentical HSCT for treating hematologic malignancies [63].

Transplantation using haploidentical related donors has also recently demonstrated encouraging results in adults with SCD (Table 4). A fludarabine/cyclophosphamide/total body irradiation (TBI) 200 cGy/ATG nonmyeloablative regimen with PTCy developed at Johns Hopkins University was first applied to 14 patients with SCD (median age: 30 years) [64]. In this study, the haploidentical regimen was well tolerated, with no observed transplant-related mortality or GVHD, but only 53% of patients achieved stable engraftment. Modifications to this regimen, including an increased dose of TBI to 300 cGy and using peripheral mobilized stem cells [65] or the addition of thiotepa [66] have improved the rates of stable engraftment. Another approach that has led to a high rate of stable engraftment in four children and young adults is the addition of two cycles of pre-HSCT immunosuppression using fludarabine and dexamethasone [67] To reduce the chemotherapy exposure in SCD adults with more advanced organ damage, the alemtuzumab/TBI regimen has been utilized as the conditioning regimen with PTCy using haploidentical donors [68]. SCD adults with more severe organ damage tolerated this regimen relatively well, but stable engraftment was observed in only 50% of patients; these patients continue on immunosuppression.

## 4. Evaluating Risks versus Benefits of Allogeneic HSCT in Adults with Sickle Cell Disease 

Balancing the risks versus benefits for allogeneic HSCT can be challenging in a nonmalignant hematologic disorder with an unpredictable clinical course, such as SCD. The only way to prove the efficacy of allogeneic HSCT over the currently available therapies in SCD will be a prospective, randomized trial comparing HSCT with standard-of-care treatment. 

The Hematopoietic Cell Transplant Comorbidity Index (HCT-CI) incorporates 17 comorbidities into a weight score that is used to predict relapse and overall mortality after HSCT [69] An intermediate or high score has been observed in 77% of SCD adults undergoing HSCT using a nonmyeloablative conditioning regimen [58] and in 30% using a reduced-intensity conditioning regimen [50]. Although this tool is useful in several malignant and nonmalignant hematologic conditions, an increasing HCT-CI score was not associated with lower survival when it was applied to patients with hemoglobinopathies [70]. This may be, in part, because this scoring system assigns three points for a total bilirubin > 1.5 times the upper limit of normal as an indicator of liver dysfunction. Because SCD is a chronic hemolytic disorder, the majority of patients will have an elevated total bilirubin, making this risk factor nondiscriminatory for HSCT outcome risk.

Understanding the risks that SCD patients are willing to accept is also important in the HSCT decision-making process. In a survey of 100 adults with SCD, 63% were willing to accept a 5% risk of HSCT-related mortality [71]. Interestingly, no significant differences in patient or disease characteristics were observed between those willing versus not willing to accept this level of risk. In a more recent survey of adults with SCD, 62% were willing to accept a HSCT-related mortality risk of 10% or greater, and 64% were willing to accept a risk of graft failure of 10% or greater [72]. In contrast, the majority (80%) were not willing to accept a risk of chronic GVHD. Findings from this survey are consistent with what we have observed in clinical practice [58]. Of 61 patients referred to our blood and marrow transplant clinic, 52 (87%) were interested in proceeding with workup for allogeneic HSCT using the alemtuzumab/TBI regimen. Based on these experiences, a large proportion of SCD adults are willing to accept the risk of HSCT-related mortality or graft failure.

While most of the clinical data on the benefits of HSCT have focused on pediatric cohorts, more recent studies have demonstrated similar improvements in acute and chronic SCD-related complications, as well as in HRQOL and healthcare utilization.

### 4.1. Acute Complications after HSCT

Several of the recent HSCT regimens in adults have demonstrated improvements in rates of VOC. In the reduced-intensity conditioning regimen developed by Ozdogu et al., the proportion of patients with ≥2 VOC/year improved from 90% pre-HSCT to no patients experiencing VOC after HSCT [50]. In the NIH cohort, pain-related admissions improved from a median of three/year pre-HSCT to zero/year at one year post-HSCT in those with stable long-term engraftment using the alemtuzumab/TBI regimen [73]. Improvements in the rates of VOC from a median of four/year pre-HSCT to a median of two/year at one year post-HSCT and to zero/year at two years post-HSCT have also been observed using the alemtuzumab/TBI regimen in the UIC cohort [74]. In this cohort, rates of acute chest syndrome also improved from 31% in the year prior to HSCT to 6% in the first year post-HSCT and 0% by the second year post-HSCT. In parallel with the improvement in VOC crises, opioid use has also gradually improved. In the study by Ozdogu et al., opioid use improved from 85% pre-HSCT to 10% at day +100, 5% at day +180, and 0% at one year post-HSCT in this study [50]. The NIH also demonstrated a gradual improvement in opioid use after HSCT [57]. 

Low rates of stroke recurrence and stable brain MRI imaging findings have been consistently demonstrated in several different HSCT approaches (Table 5). In a combined analysis of six different studies for HSCT in adults, only one stroke recurrence out of 33 patients (3%) with stable engraftment has been observed [35,57,58,64,65,66]. This rate of stroke recurrence is substantially lower than what is observed with chronic red blood cell transfusion therapy (18%). Furthermore, stable brain MRI and angiography findings have been demonstrated in those patients that had pre- and post-HSCT brain imaging studies and stable engraftment [35,57]. 

### 4.2. Cardiovascular Function after HSCT

In two separate studies using the alemtuzumab/TBI nonmyeloablative conditioning regimen, improvements in the TRJV, a strong predictor for mortality, have been observed from pre-HSCT values to values at one or two years post-HSCT (Figure 1) [57,58]. In the STRIDE study, two SCD patients with a TRJV ≥ 2.7 m/s pre-HSCT had normalization of their TRJV, while one patient with a TRJV = 2.4 m/s pre-HSCT had an increase to 2.7 m/s at one year post-HSCT [35]. An improvement in BNP, a marker of cardiac myocyte stress, has also been observed in SCD adults from median values pre-HSCT (21 pg/mL, range: 5–92 pg/mL) to one year post-HSCT (9 pg/mL, range: 1–26 pg/mL) [58]. Right heart catheter-defined pulmonary hypertension is a strong predictor of early mortality in adults with SCD [75]. One report of a 32-year-old woman with SCD and pulmonary hypertension requiring supplemental oxygen demonstrated a dramatic improvement after haploidentical HSCT [76]. This patient’s TRJV improved from 2.8 m/s pre-HSCT to 2.4 m/s at two years post-HSCT, with normalization of her pulmonary artery pressures and being completely weaned off supplemental oxygen.

### 4.3. Health-Related Quality of Life after HSCT

Until recently, the benefits of HSCT on HRQOL were not as clear in adults undergoing HSCT. Pediatric cohorts using primarily myeloablative regimens have demonstrated that HRQOL may initially decline within the first three months post-HSCT, followed by an improvement at one year post-HSCT compared to pre-HSCT scores [77,78]. More current experience with reduced-intensity and nonmyeloablative regimens have demonstrated that adults with SCD, who have more chronic SCD-related complications, still see a benefit to their HRQOL after allogeneic HSCT (Figure 2). In adults who underwent HSCT using the alemtuzumab/TBI regimen, improvements in bodily pain, general health, vitality, and social function scores were observed from pre-HSCT to one year post-HSCT according to the short form (SF)-36 HRQOL questionnaire [58]. In this study, both bodily pain and general health progressively improved starting as early as day +30 post-HSCT. Furthermore, the SF-6D summary score, which provides an overall utility score based on societal weights derived from items in the SF-36 questionnaire, demonstrated clinically meaningful improvements in HRQOL starting at day +30 and progressively increasing at day + 90 and one year post-HSCT. Improvements at one year post-HSCT have also been demonstrated in adults undergoing reduced-intensity conditioning regimens. In the study by Ozdogu et al., significant improvements in the SF-36 score were observed for role physical, general health, and bodily domains while trends in improvement were observed for energy/vitality, social functioning, role emotional, and mental health from pre-HSCT to one year post-HSCT [50]. In the STRIDE study, improvements in pain interference were observed at six months and one year post-HSCT, while physical function was improved at one year post-HSCT compared to pre-HSCT values using the PROMIS-57 profile [35].

### 4.4. Healthcare Utilization after HSCT

Adults require more healthcare, determined by emergency room visits, frequency of inpatient hospitalization, and inpatient hospital days, than children with SCD [32]. Two studies using the alemtuzumab/TBI regimen demonstrated that HSCT in adults reduced healthcare utilization (Figure 3). In the NIH cohort, the rate of hospitalizations progressively improved from pre-HSCT to one year and two years post-HSCT. In adults treated at the University of Illinois at Chicago, there was a trend for an increase in the median number of inpatient hospital days from the year pre-HSCT (22 days/year) to the year of the HSCT (38 days/year), which included the number of days hospitalized for the HSCT procedure [74]. By the second year post-HSCT, the median number of inpatient hospital days (one day/year) was significantly lower than pre-HSCT values. Median rates of emergency room visits were significantly lower at both one year (two visits/year) and two years (one visit/year) post-HSCT compared to the year pre-HSCT (four visits/year).

## 5. Future Directions

Current guidelines in symptomatic children with SCD advocate for HSCT as early as possible if an HLA-matched related donor is available [79]. As safer and more effective conditioning regimens are emerging, guidelines and recommendations will need to be updated for when to consider HSCT in adults with SCD. Current data suggest that nonmyeloablative transplants from either matched-related or haploidentical donors may represent relatively safe options for adult patients with advanced SCD. Prediction models, based on clinical complications [80]. Genetic risk modifiers [81], or gene expression patterns [82], may help identify high-risk SCD patients for HLA-matched and alternative donor HSCT, but will require validation in future studies. The long-term risks and benefits of HSCT, such as on fertility and overall survival, will also need to be better understood, particularly in the context of emerging therapies, such as selectin inhibitors (crizanlizumab), small-molecule inhibitors of sickle hemoglobin polymerization (voxelotor), and gene therapy to treat SCD. 

## Figures and Tables

**Figure 1 jcm-08-01565-f001:**
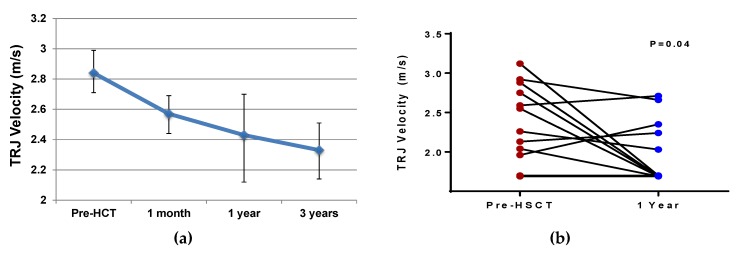
Improvements in the tricuspid regurgitant jet velocity (TRJV) after hematopoietic stem cell transplantation in adults with sickle cell disease from (**A**) the National Institutes of Health (*n* = 13) and (**B**) University of Illinois at Chicago (pre-HSCT mean = 2.3 ± 0.5 m/s, post-HSCT mean = 2.0 ± 0.4 m/s) (*n* = 12) cohorts, using an alemtuzumab and total body irradiation conditioning regimen.

**Figure 2 jcm-08-01565-f002:**
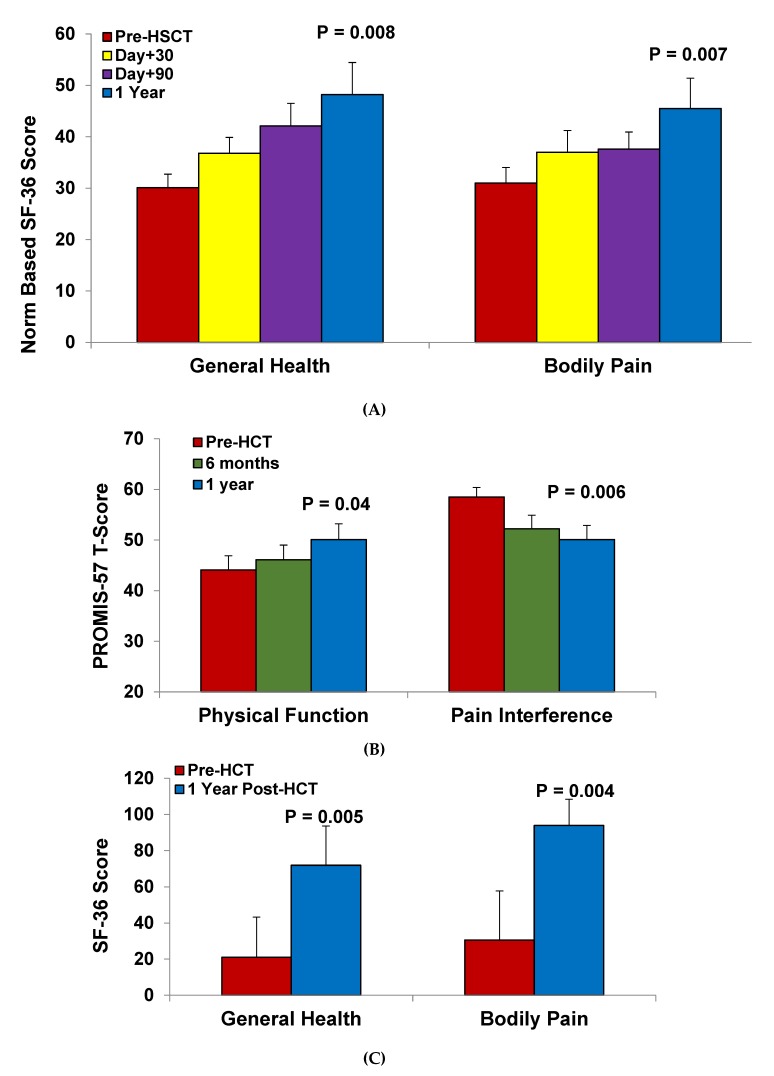
Increased health-related quality of life after hematopoietic stem cell transplantation in adults with sickle cell disease from (**A**) the University of Illinois (*n* = 9), (**B**) University of Baskent (*n* = 20), and (**C**) the multi-center STRIDE study (*n* = 17).

**Figure 3 jcm-08-01565-f003:**
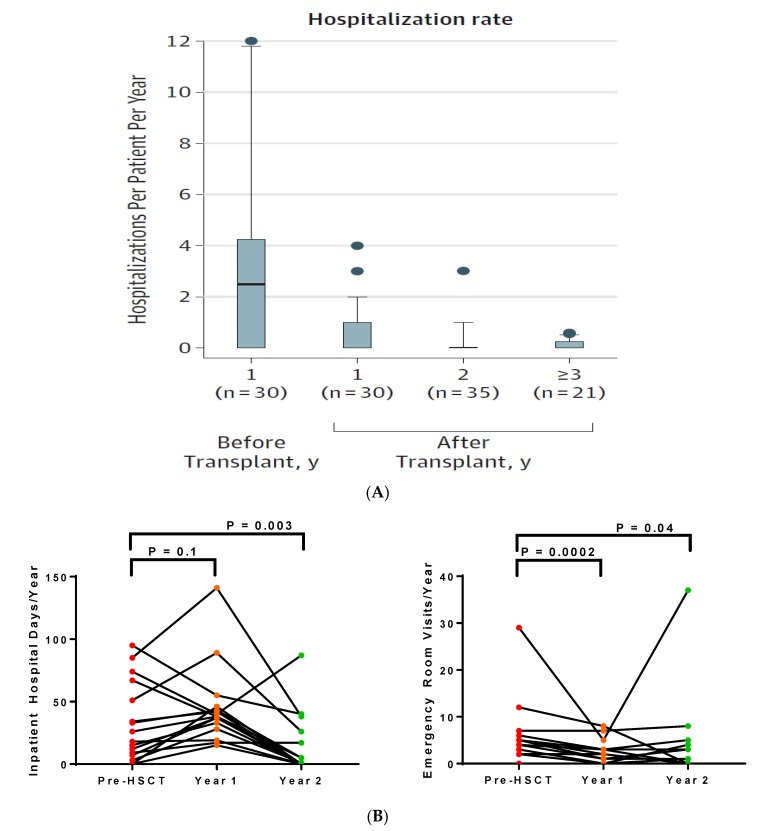
Lower healthcare utilization in adults with sickle cell disease after allogeneic hematopoietic stem cell transplantation from (**A**) the National Institutes of Health and (**B**) the University of Illinois at Chicago cohorts, using an alemtuzumab and total body irradiation conditioning regimen.

**Table 1 jcm-08-01565-t001:** Indications for allogeneic hematopoietic stem cell transplantation in sickle cell disease [34,35,36,37].

Indication		HLA-Matched Related Donor	Alternative Donor*
**Predictors for Increased Mortality**	≥2 VOC/year	X	X
	Recurrent acute chest syndrome	X	X
	Stroke	X	X
	Cognitive impairment + Abnormal cerebral MRI	X	X
	TRJV ≥ 2.7 m/s	X	X
	Sickle nephropathy	X	
**Predictors for Increased Morbidity**	Red blood cell alloimmunization	X	
	Recurrent priapism	X	
	Osteonecrosis of multiple joints	X	

HLA, human leukocyte antigen; VOC, vaso-occlusive crisis; MRI, magnetic resonance imaging: TRJV, tricuspid regurgitant jet velocity. * Despite hydroxyurea and/or chronic red blood cell transfusion therapy.

**Table 2 jcm-08-01565-t002:** Reduced-intensity conditioning regimens in adults with SCD.

Conditioning Regimen	GVHD Prophylaxis	*N*	Age (years)	Graft Source	Event-free Survival	Overall Survival	aGVHD (≥grade 2)	cGVHD (≥moderate)	Reference
Fludarabine (150 mg/kg) Busulfan (3.2 mg/kg) Cy (29 mg/kg) ATG (15 mg/kg) Total body irradiation (200 cGy)	Cy 33 mg/kg/day on day +3, 4 Sirolimus	20	20–45	20 MRD	100%	100%	5%	0%	[50]
Fludarabine (175 mg/m^2^) Busulfan (13.2 mg/kg) ATG (6 mg/kg)	MTX 7.5 mg/m^2^/day on day +1, 3, 6, 11 Cyclosporine or tacrolimus	22	17–36	17 MRD 5 MUD	82%	91%	18%	14%	[35]

SCD, sickle cell disease; aGVHD, acute graft versus host disease; cGVHD, chronic graft versus host disease; Cy, cyclophosphamide; ATG, antithymocyte globulin; cGy, centigray; MRD, matched related donors; MUD, matched unrelated donors.

**Table 3 jcm-08-01565-t003:** Nonmyeloablative alemtuzumab/TBI conditioning regimens in adults with SCD.

Conditioning Regimen	GVHD Prophylaxis	*N*	Age (years)	Graft Source	Event-free Survival	Overall Survival	aGVHD (≥grade 2)	cGVHD (≥moderate)	Reference
Alemtuzumab 1 mg/kg Total body irradiation (300 cGy)	Sirolimus	30	17–65	MRD	87%	97%	0%	0%	[57]
		13	17–40		92%	100%	0%	0%	[58]
		17	14–39		89%	100%	0%	0%	[59]

**Table 4 jcm-08-01565-t004:** Haploidentical hematopoietic stem cell transplantation (HSCT) in adults with SCD.

Conditioning Regimen	GVHD Prophylaxis	*N*	Age (years)	Graft Source	Event-free Survival	Overall Survival	aGVHD (≥grade2)	cGVHD (≥moderate)	Reference
Fludarabine (150 mg/m^2^) Cy (29 mg/kg) ATG (4.5 mg/kg) Total body irradiation (200 cGy)	Cy 50 mg/kg/day on day +3, 4 MMF until day +35 Tacrolimus or Sirolimus	14	15–42	Bone Marrow	57%	100%	0%	0%	[64]
Alemtuzumab (1 mg/kg) Total body irradiation (400 cGy)	Cy 50 mg/kg/day on day +3, 4 Sirolimus	12	20–56	PBSC	50%	92%	0%	0%	[68]
Fludarabine (150 mg/m^2^) Cy (29 mg/kg) ATG (4.5 mg/kg) Total body irradiation (300 cGy)	Cy 50 mg/kg/day on day +3, 4 MMF until day +35 Sirolimus	8	20–38	PBSC	75%	88%	25%	13%	[65]
Pre-HSCT Immunosuppression: Fludarabine (200 mg/m^2^) Dexamethasone (125 mg/m^2^) Conditioning: Fludarabine (210 mg/m^2^) Busulfan (520 mg/m^2^) ATG (4.5 mg/kg)	Cy 50 mg/kg/day on day +3, 4 MMF until day +28 Tacrolimus	4	12–24	Bone Marrow (*n* = 3) PBSC (*n* = 1)	100%	100%	0%	0%	[67]
Fludarabine (150 mg/m^2^) Cy (29 mg/kg) Thiotepa (10 mg/kg) ATG (4.5 mg/kg) Total body irradiation (300 cGy)	Cy 50 mg/kg/day on day +3, 4 MMF until day +35 Sirolimus	15	7–40	Bone Marrow	93%	100%	13%	0%	[66]

**Table 5 jcm-08-01565-t005:** Stroke recurrence and brain imaging findings in adults with sickle cell disease after allogeneic hematopoietic stem cell transplantation (HSCT).

HSCT Type	Stroke Recurrence	Brain MRI Findings	Reference
• Nonmyeloablative, • Matched-related donor	0 out of 9	Stable brain MRI and angiography findings	[57]
• Nonmyeloablative• Haploidentical related donor	0 out of 3	–	[64]
• Nonmyeloablative• Matched-related donor	1 out of 4	–	[58]
• Nonmyeloablative• Haploidentical	0 out of 2	–	[65]
• Nonmyeloablative• Haploidentical	0 out of 9	–	[66]
• Reduced-intensity• Matched-related and unrelated donor	0 out of 2	Stable brain MRI in 17 patients	[35]
**TOTAL**	**1 out of 29 (3%)**

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
