# Peer review of "Allogeneic Hematopoietic Stem Cell Transplantation for Adults with Sickle Cell Disease"

_jcm, 2019, doi:10.3390/jcm8101565_

Round 1

Reviewer 1 Report

In this manuscript, the authors provide a thoughtful, comprehensive review of allogeneic transplantation for adults with sickle cell disease. 

-The authors should provide a reference for Table 1 regarding where these indications for transplantation are reported. Further, Sickle nephropathy is listed under predictors for increased morbidity, but it has been shown by multiple investigators to be associated with an increased risk for mortality.

-The authors suggest in the Table that indications for Alternative Donor transplant should be considered only despite hydroxyurea and/or chronic red blood cell transfusion therapy, but this is not mentioned for HLA-Matched Related Donor transplant. This should be discussed in the text.

-The reference numbers in the Tables should be corrected

-The NIH group wrote a manuscript about acute and chronic pain before and after HLA-matched sib HSCT (Darbari et al. BJH, 2019). Consider referencing the article in the section describing Acute Complications after HSCT.

-P-values would be helpful to add to Figure 2 and 3A.

-Under Future Directions, the authors state that prediction models based on clinical complications, genetic risk modifiers, or gene expression patterns will help identify high risk SCD patients for HSCT. Are we ready at this time to incorporate those models into our donor selection practices?  This should be discussed more in the Future Directions or Indications for HSCT section.

Author Response

Dear Reviewers,

Thank you for providing the comments and suggestions, which have helped strengthen this manuscript.  Please see below point-by-point answers addressing each comment.

1) The authors should provide a reference for Table 1 regarding where these indications for transplantation are reported. Further, Sickle nephropathy is listed under predictors for increased morbidity, but it has been shown by multiple investigators to be associated with an increased risk for mortality.

As suggested, we have added reference #34 - 37 to the title of Table 1.  We have also moved sickle cell nephropathy from the morbidity to mortality predictors.

2) The authors suggest in the Table that indications for Alternative Donor transplant should be considered only despite hydroxyurea and/or chronic red blood cell transfusion therapy, but this is not mentioned for HLA-Matched Related Donor transplant. This should be discussed in the text.

We have now included that alternative donor transplant should be considered after trying hydroxyurea or red blood cell transfusion therapy to the text (page 3, lines 109-112). 

3) The reference numbers in the Tables should be corrected

Thank you for pointing out this error and the references throughout the tables have been corrected

4) The NIH group wrote a manuscript about acute and chronic pain before and after HLA-matched sib HSCT (Darbari et al. BJH, 2019). Consider referencing the article in the section describing Acute Complications after HSCT.

Thank you for providing this reference and findings from this publication have been added to this manuscript (page 12, lines 268 - 270).

5) P-values would be helpful to add to Figure 2 and 3A.

In agreement, we have added P-values to Figures 2A, B, and C.  

6) Under Future Directions, the authors state that prediction models based on clinical complications, genetic risk modifiers, or gene expression patterns will help identify high risk SCD patients for HSCT. Are we ready at this time to incorporate those models into our donor selection practices?  This should be discussed more in the Future Directions or Indications for HSCT section.

We have modified this sentence accordingly and stated that these prediction models will require validation in future studies (page 18, lines 373, 374). 

Reviewer 2 Report

This is a well motivated review in the light of improvments in HSCT ,which motivates this therapy in selected adult patients with Sickle cell disease.The review is comprehensive and easy to read.

The only way to prove efficacy is a prospective randomized trial comparing HSCT with standard of care.This option may be discussed with pros and cons.

Minor.In Fig 1A and Fig 2 the number of patients included should be given.

In fig 1B Mean values may be given.

Refrence 1 Please proide volume and pages.

Author Response

Dear Reviewer 2,

Thank you for your comments and suggestions which have helped strengthen this manuscript.  Please see below a point-by-point reply to the comments.

1) The only way to prove efficacy is a prospective randomized trial comparing HSCT with standard of care.This option may be discussed with pros and cons.

In agreement, we have added to the section evaluating risks versus benefits that a prospective randomized study will be required to prove efficacy of HSCT over standard of care (page 12, lines 237 - 239).

2) In Fig 1A and Fig 2 the number of patients included should be given.

We have included the number of patients to the figure legends for Figures 1 and 2.

3) In fig 1B Mean values may be given.

We have added the mean +/- standard error for figure 1B in the figure legend.

4) Reference 1 Please provide volume and pages.

Thank you for picking up this error - the reference has been corrected and now includes the volume and pages.